# The Relationship between Physiotherapist and Patient: A Qualitative Study on Physiotherapists’ Representations on This Theme

**DOI:** 10.3390/healthcare10112123

**Published:** 2022-10-25

**Authors:** Silvia Monaco, Alessia Renzi, Beatrice Galluzzi, Rachele Mariani, Michela Di Trani

**Affiliations:** Department of Dynamic and Clinical Psychology and Health Studies Sapienza, University of Rome, 00185 Rome, Italy

**Keywords:** physiotherapist, professional–patient relationship, health professional, emotional text mining, qualitative study

## Abstract

The physiotherapist represents a resource for the psychophysical well-being of an individual. Specific characteristics of the physiotherapist–patient relationship can influence the outcome of rehabilitation. This study aimed to explore physiotherapists’ representations on how they perceive their relationship with their patients, in order to highlight helpful elements in promoting the outcome of the intervention. In this study, 50 physiotherapists (27 females and 23 males; mean age = 42 years; sd = 12.2) participated in an individual interview, conducted remotely via videocall. Socio-demographic and occupational data were collected. The interviews were recorded and transcribed. Texts were analyzed using emotional text mining (ETM). Participants organized their work by means of three categories: (1) work with the patient, in which the physiotherapists described two complementary elements of the therapy, which are the observable-technical aspects of their work and the internal predispositions; (2) the healing process, highlighting the aims of their intervention, including the physical pathology and the relationship with the patients; (3) physiotherapist as a psychologist, describing the attempt to understand patients’ emotional experience to gradually transition to the practical intervention. Understanding the emotional and relational processes that form the basis of physiotherapist practice can contribute to the development of interventions in which the body and the mind can be integrated, resulting in a real person-centered point of view.

## 1. Introduction

The physiotherapist (PT), as well as other healthcare professionals, represent a resource for the psychophysical well-being of patients. The spectrum of action of this profession appears to be particularly broad, as PTs work with numerous types of patients in different contexts. In the healthcare context, physiotherapy is aimed to support the patient by recovering a lost or damaged motor function. Biopsychosocial literature [1] underlines the importance of the quality of the relationship between PT and patient in order to reach a complete recovery. In fact, the physical-motor rehabilitation seems to also have a positive effect on both cognitive and psychological functions in a patient. In light of this, an interesting study [2] showed that the physiotherapy rehabilitation of the elderly contributed to their improvement in physical functioning and endurance, as well as to their enhancement of cognitive functioning and psychological well-being, thereby promoting a significant reduction in depression and anxiety symptoms. A meta-analysis investigating the effect of physiotherapy in older adults with depression found that physiotherapy improves physical health, body image, patients’ coping strategies, quality of life, and independence in daily activities [3]. Vancampfort et al. [4] showed the positive effect of the use of the 2 min walk test as an integral part of the treatment of patients suffering from depression.

As the studies focused on the PT–patient relationship, several investigations have explored the possible influence of this relationship on the outcome of the physiotherapy. In this direction, one study [5] examined PTs’ opinion about what are the essential factors of the PT–relationship for promoting a positive treatment outcome. These factors were divided into two main categories: “prerequisites dimension” (related to life experiences, personal characteristics, level of education, and practical skills), and “interaction dimension” (related to fostering and maintaining contact with the patients, the elements of the therapeutic process, and the structural aspects of treatment). PTs reported to be concerned about creating a climate of positive affection in the therapy room; recognizing the importance of constructing an empathic, respectful, sensitive, involved dialogue with patients; as well as encouraging independence in patients to seek their own personal resources for the healing process. Klaber et al. [6] reported that encouraging patients’ self-efficacy and promoting positive coping strategies showed a positive effect on patients’ compliance levels, positively influencing the rehabilitation outcome.

Over time, studies aiming to improve the effectiveness of physiotherapy treatment included the investigation of several mediators such as communication; psychological interactions between patients and clinicians; and, more specifically, the therapeutic alliance. The term “therapeutic alliance” refers to the interpersonal relationships formed between a therapist and a client through cooperation, dialogue, therapist empathy, and respect [7,8], which may operate in conjunction with or independently of certain treatments. The majority of the existing research supporting the therapeutic alliance’s beneficial role in improving treatment outcomes in medical diseases comes from the literature on psychotherapy and medicine. Babatunde et al. [8], in their scoping review, found that the therapeutic alliance has been poorly studied in physiotherapy rehabilitation literature, finding that it can enhance exercise adherence. They came to the conclusion that more research is required to fully explore this issue, particularly from the perspective of therapists, in order to improve physical rehabilitation programs. In this light, Unsgaard-Tøndel and Søderstrøm [9] showed that PTs indicate four key elements for establishing a good therapeutic alliance with patients: (1) presence and empathy within a biopsychosocial perspective as central to building the therapeutic alliance; (2) active listening and the adaptation of the treatment to the specific patient; (3) clinical experience, which is important to integrate the psychological and social domains into physiotherapy management; and (4) the use of sensitive communications to assist patients in acquiring new insights about their possibilities. In the same direction, Brunner et al. [10] found that PTs’ higher confidence in managing their patients was associated with higher patient-reported working alliances. On the contrary, the review of Taccolini Manzoni, et al. [11] concluded that the international literature did not provide sufficient evidence of a strong relationship between the therapeutic alliance and physiotherapy outcome. In conclusion, it seems possible to sustain that the PT is a valuable professional within a multidisciplinary team aimed to treat patients affected by both physical and mental pathology. Furthermore, although the literature seems to highlight that some characteristics of the PT–patient relationship can influence the rehabilitation outcome, to the best of the authors’ knowledge, no prior studies have explored how PTs perceive their work and their relationship with patients. The analysis of PTs’ representations on these themes could be important in underlining specific risk and protective features, with the broader aim to realize future psycho-educational intervention aimed to improve physiotherapy treatment outcomes.

Therefore, the principal aim of the present study is to explore, through interviews, how PTs think about their work and their relationship with patients through the application of a computerized text analysis program. Furthermore, the possible association between PTs’ socio-demographic dimensions with specific elements of the PT–patients relationship will be explored.

## 2. Materials and Methods

This study was carried out in accordance with the code of ethics of the World Medical Association (Declaration of Helsinki) for experiments involving humans. Ethical approval was granted by the ethics committee of the Department of Dynamic and Clinical Psychology and Health Studies of Sapienza. The data collection took place during the period of October 2021 to December 2021.

### 2.1. Participants

The study focused on a convenient sample of PTs (mainly residents of Rome and its surroundings) enrolled using the following inclusion criteria:(1)exercising their profession as a PT;(2)being active at work at the time of participation in the study;(3)working as a PT for at least two years, considering two years the minimum needed to collect information from professionals with working experience also before the COVID-19 pandemic outbreak.

In total, 50 PTs (see Table 1) working in public and private contexts participated in the study: 27 females and 23 males with a mean age of 42 years (SD = 12.2). The mean of participants’ work experience was 18 years, with an SD of 12 years. The majority of participants work in structures affiliated to the National Health System (60%). Participants reported to work mainly with both orthopedic (86%) and neurological (62%) patients.

### 2.2. Procedure

The psychologist responsible for the research protocol contacted the PTs by telephone to explain the study. More specifically, the psychologist informed the PTs that we were conducting a study aimed to collect personal considerations about the relationship between PT and their patients through a telematics interview. Then, the psychologist arranged an appointment for the interview and the sociodemographic and occupational questionnaire with each PT that voluntarily agreed to participate. Each participant signed an informed consent form and a privacy policy form before data collection. The individual meetings took place remotely via videocall and participants were asked to carry out the interview in a quiet, reserved, and comfortable place. The choice of remote interviews was because of compliance with the rules for containing the spread of COVID-19. All interviews were conducted by two psychologists with experience in the conduction of interviews. All interviews were audio recorded, transcribed verbatim, and the texts were analyzed using T-LAB software [12].

### 2.3. Measures and Data Analusis

#### 2.3.1. Socio-Demographic and Occupational Questionnaires

Aimed to collect data regarding gender, age, years of work experience, workplace (structures affiliated to Health National System, structures not associated to Health National System, private practice, cooperative for home-intervention assistance), and the population of patients treated with the option to report more categories of patients. This questionnaire was completed before the interview.

#### 2.3.2. Interview

The technique of the interview is the core of this study, specifically the open-ended interview style. The interview was characterized by a single open-ended question: “Can you tell me about the relationship with your patients?” [“Può parlarmi di come vive la relazione con i suoi pazienti?”]. This particular form represents a flexible and unstandardized approach called “free association”, in which participants have to connect their thoughts to the original theme [13]. In this way, the participant is free to associate different events, concepts, or personal memories to the original question, promoting the materialization of related representations.

In order to identify PTs’ representations about the relationship they have with their patients, we applied the emotional text mining methodology (ETM) [14]. Such a methodology of natural language processing (NPL) is based on a socio-constructivist approach that, based on the association of words, allows to identify the general themes and symbolic-cultural categories of the construction of meaning. This methodology is composed of a structured and reproducible procedure and has already been used by several authors in the field of sociology [15], health [16], and psychology [17,18]. Moreover, it has already been applied to study patients’ experiences in a patient-centered view [19]. It allows for the study of emotional dimensions that organize a relationship, starting from the use of language, and considering the semiotic and semantic levels. ETM follows five main phases: cleaning of the final corpus, with the selection of final keywords; calculating three lexical indexes to assess the richness of the corpus and the possibility to statistically process data (token: number of words; hapax percentage: the percentage of all single words, which occurs only one time in the text, it should be around the 50%; type/token ratio: an index of lexical diversity, it should be <1.20) [20]; performing a cluster analysis with a bisecting k-means algorithm on the text-keyword matrix [21], assessing the intra-class correlation coefficient to choose the optimal solution; completing a correspondence analysis on the term-cluster matrix [22]; and, finally, interpreting the results. To deepen the validity and the reliability of this methodology, please see [14,20,21,22]. For the final interpretation of results, four principal elements must been taken into account: (1) the word’s list of each factor, taking into consideration the polarization and the absolute contribution of each word to their factor—words with the highest percentage of absolute contribution are more significant and iconic of the dimension carried from that factor; (2) the word’s list of each cluster in order of importance by their context units percentage; (3) the collocation of the cluster among the factors to define the factorial dimension in which they are placed; and (4) a selection of elementary context (interview’s fragments) sorted by weight. This last point refers to an analysis made by T-LAB, where a score is applied connected to the relevance to each document present in the cluster, the TF-IDF of Salton [23].

### 2.4. Statistical Analysis

Using the Statistical Package for Social Science version 25 for Windows (SPSS version 25; IBM, Armonk, NY, USA), we performed a Chi-square test on the final partition of the cluster, following the collection of socio-demographic variables. This way, we can identify the significant presence of a variable in a cluster. *p* < 0.05 was considered significant.

## 3. Results

The transcription verbatim of all 50 interviews created a whole corpus of medium dimension (token: 74763). The lexical indexes of the final corpus (TTR: 0.089; HAPAX%: 52%) assessed its richness and allowed us to continue the analysis according to ETM methodology. The keywords selected during analysis allowed for the classification of 99.11% of the content units. The correspondence analysis and the clustering validation measures showed that the optimal solution was three factors and four clusters.

The PTs symbolize the relationship with the patient through three main categories: (1) work with the patient, (2) the healing process, and (3) PT as a psychologist. Each factor represents two contrapose polarities, where the following cluster will be positioned.

### 3.1. Factor 1—Work with the Patient

The first factor (Table 2) represents the work with the patient carried out by the PT, composed of two complementary elements of therapy: the visible PT–patient relationship and the PT’s internal predisposition. “The visible relationship, PT–patient” refers to the observable aspects of the PT–patient relationship, which can be seen from words related to the more “technical” side of this job, such as rehabilitation or pathology, combined with those that refer to the formation of the therapeutic alliance and the goal of the therapy, such as trusting, bonding, creating, and changing. On the other hand, “internal predisposition” refers to the whole set of actions preparing the PT to face the work with the patient in a comfortable way. In fact, we can find a lot of verbs that express the act of creating a connection with the patient: to talk, to understand, to see, to feel, to search for, and to help. In the poles, some elements of the therapeutic alliance also emerge, but they belong more to the PT than to the PT–patient couple. In fact, the words mostly concern the actions implemented by the PT to facilitate the creation of the alliance, which are partly directly observable by the patient (for example, to talk and to help) and partly manifested in the PT–patient relationship in a less visible way (for example, to search for, to feel, to see, and to understand).

### 3.2. Factor 2—The Healing Process

The second factor (Table 3) refers to “the healing process”, which is expressed through two poles: pathology and relationship. The pole named “pathology” refers to the object of the physiotherapy intervention as shown by words like problem, change, orthopedic, and neurological. In contrast, the pole “relationship” refers to the relationship between the PT and patient. This relationship appears to be rich, with aspects related to the profession, as well as psychological and emotional aspects that belong to the members of therapeutic couple. This can be seen in words such as to create, to rehabilitate (which has a very different meaning from ‘to change’), to trust, to think, and in_ mind. Furthermore, it clearly emerges how these two poles can be framed within well-defined models of physiotherapy: “pathology”, in fact, mostly belongs to a “medical model” based on repairing a physical damage, far from a theoretical framework that takes into consideration the psychological and social aspects. It is indeed interesting that this model is defined by some authors as an “orthopedic model of medicine” [24], referring to the mechanical action of adjusting an injured part of the body that concerns precisely the most practical aspect of the PT’s work. On the contrary, the pole “relationship” seems to be inscribed within a “care model”, which is closer to the psychological field and the biopsychosocial theoretical framework of healthcare. Compared with this pole, the word “think” and the compound form “in_mind” are particularly interesting because they reflect on the mental space created by the PT to let the patient enter into his representative world and the way in which they can keep and hold it in their own mind, in a similar manner to the parental holding function described by Winnicott [25]. As will be better explained later through the description of the clusters, the work of the PT cannot be fully described and understood if only the technical side of the relationship is taken into consideration.

### 3.3. Factor 3—Physiotherapist as a Psychologist

The third factor (Table 4), “physiotherapist as a psychologist”, is composed of two polarities: comprehension and action. The pole named “comprehension” represents the attempt to understand the patient’s experience regarding both the illness and general life, highlighting a type of understanding that is both cognitive and emotional. This can be seen from the use of perceptual verbs, such as to see and to feel, as well as from words that refer to entering contact with the other and “discovering him” (to find, to talk, to tell, interesting). On the other hand, the pole named “action” describes a gradual transition from the understanding phase to the practical intervention of the PT. This is shown by the presence of verbs that still refer to the sphere of comprehension (to understand, to listen), but also words related to the function of the support given from the PT and what the PT can do for the patient.

In this vein, the PT (“as a psychologist”) seems to be reasoning on the modalities that can be used to help patients overcome the difficulties going beyond the physical aspect of the disease.

### 3.4. Clusters

The interpretation of the factorial space highlights the symbolic categories by which PTs emotionally categorize their relationship with their patients and supports the cluster interpretation. In Table 5, it is possible to see how the four clusters that emerged from the analysis are positioned based on the polarities of each factor.

As shown in Table 6, participants represented this relationship through four main categories: request of integration (26.7% context units, UC), the adequate professional (22.2% UC), experiences of the relationship’s elaboration (27.3% UC), and the “not-adequate” professional (23.8% UC).

In the first cluster (Table 6), called “request of Integration”, the interviewees described their relationship with their patients, highlighting both the relational aspects and the principal objective of the physiotherapy intervention, which is to help the patient. This cluster seems to reflect PTs’ need to overcome the dichotomy between person and body, as well as between relational elements (emotions, to listen, to understand) and technical elements (method, to search for, to be able to), in order to practice an integrated model of care that cannot fail to consider the psychological and emotional aspects of patients. Below is a sample excerpt of the interviews:

<<… So, I try to understand as much as possible on an emotional level, but there are faster things, maybe they do three or four therapies, they resolve, they go away, so the approach is completely different when you have that patient. But I always emh… always try to listen and help the person in front of me as much as I can on an emotional and psychological level>>. […Quindi io cerco di capire il_più_possibile a livello emotivo però lì sono cose più veloci, magari fanno tre quattro terapie, risolvono, se ne vanno quindi è tutto diverso l’approccio quando hai quel paziente. Però io comunque sempre ehm… cerco sempre di ascoltare e di aiutare quanto posso la persona che ho davanti a livello emotivo e psicologico.] (Score: 427.929)

The second cluster, “the adequate professional”, describes PTs’ working experience that takes into account both the professional side (patient, pathology, problem, to change, to work, and job) and patients’ emotional implications of illness (different, family, psychology, and psychologist). This highlights the importance of focusing on the pathology, but also on the broader consequences on patients’ lives and the specific context (patients’ lives) in which the pathology is inserted. In this light, the “adequacy” of their work experience is represented by a holistic consideration of patients and their problems that can shape the type of professional intervention performed.

Here, some excerpts from the interviews that are representative of this cluster are presented:

<<… there are patients who actually have other problems that are to be linked outside the pathology, perhaps they have family or work problems. Or even patients with lower back pain, general back pain, sciatica, etc. At the end, they also have psychological problems.>> […sono pazienti che poi effettivamente hanno altri problemi che che sono da da da legarsi al di fuori della patologia, hanno problemi magari familiari o lavorativi. E oppure anche pazienti con problematiche di lombalgia, schiena, sciatica eccetera, alla fine hanno anche problemi psicologici.] (Score: 320.511)

<<…it also depends on the work you have to do. Then what I do know if I work with a neurological patient, is that the approach is completely different, because maybe on a cognitive level, in short, there are dysfunctions, problems, and therefore, I treat them differently. And here, it is also from the patient, that is, from the person you are in front of, and from the problem.>> […dipende anche dal lavoro che devi fare. Poi che ne so se lavoro con un paziente neurologico l’approccio è completamente diverso, perché magari a livello cognitivo, insomma, ci sono delle disfunzioni, dei problemi, e quindi li tratto diversamente. Ed ecco pure dal paziente cioè diciamo dalla persona che ti trovi davanti e dalla problematica.] (Score: 173.063)

The third cluster, “relationship’s Elaboration”, expresses PTs’ emotional-relational experience to the specific patient that can guide the professional’s attitude towards the patient. Interestingly, while the previous cluster’s first word was “patient”, in this cluster, this term is replaced with “person”, which is a less technical word. The presence of this word, in addition to the abundance of verbs (to see, to speak, to feel, to become, to happen), could be interpreted as a PT’s attempt to elaborate the experiences of the relationship with the patient. It represents a way of assimilating events of the patient’s narration (situation, to happen) that could be dramatic and emotionally engaging for the PT.

Here, some excerpts from the interviews that are representative of this cluster are presented:

<<…And simply, those fifteen minutes that you leave, the work becomes much more rich than to work in a mode like, “I do not hear, I do not see, I do not speak, but we just work.” I repeat then, maybe this is one of mine… that is, it is one of my modalities, that if you hear another therapist tell you, “this one is doing everything wrong… that is, it is not good, it is not professional.” […E banalmente poi quei quindici minuti che ti lasci di lavoro diventano molto più proficui rispetto a lavorare in modalità tipo non sento, non vedo, non parlo però lavoriamo e basta. Ripeto poi questa magari è una mia… cioè è una… una mia modalità che se senti un altro terapista ti dice “questa ha sbagliato tutto, sta facendo … cioè non non va bene, non è professionale.”] (Score: 118.946)

<<With a thirty-five-year-old boy who had a brain aneurysm, and he is… and so he had the same stroke as the seventy-year-old man, he tells him, “oh roll up your sleeves boy! Now you need… you need to get your driver’s license back, you have to get strong, and you have to find a girl, and if your girlfriend left you, you have to find another one; you are young and strong!”>> [Con un ragazzo di trentacinque anni che ha avuto un aneurisma cerebrale e si è… e quindi ha avuto lo stesso ictus del signore di settant’anni, gli dice “oh rimboccati le maniche ragazzo! Qua bisogna di riprendere la patente, devi… devi farti forza, e ti devi trovare la ragazza, e se la ragazza t’ha lasciato te ne devi trovare un’altra, sei giovane e forte!”] (Score: 103.140)

The fourth cluster, the “not-adequate” professional, seems to express PTs’ emotional experience, both as workers and as people, also highlighting the difficulties faced (pain). Even in this cluster, the first word is patient, a technical term that seems to express the need for emotional distance, followed by terms that refer to the relational sphere (to trust, to enter, relationship, bond). All of this seems to express difficulties in the relationship with patients and the feeling of being unprepared to adhere to all of the patients’ expectations regarding the relationship. This can make the PTs feel like they are “not-adequate” professionals.

Here, some excerpts from the interviews that are representative of this cluster are presented:

<<It is not possible to create a relationship, more than anything else. Essentially this. Yes um… at least in my work situations, then here I know that those who work in the studio experience a different relationship. Um…also the relationship with problems that is… the relationship also depends a little on the problem, because the problem is that the patient then designates the duration of the therapy.>> [Non è possibile creare un rapporto più che altro. Essenzialmente questo. Sì ehm… almeno nelle mie situazioni lavorative, poi ecco so che chi lavora in studio vive un rapporto differente. Ehm anche il rapporto con problematiche cioè… il rapporto dipende anche un_pochino dalla problematica, perché la problematica che ha il paziente poi designa la durata della terapia.] (Score: 534.466)

<<… there is disappointment… then we must try to make people understand that it is… and obviously we do not do it alone because it is very difficult, but unfortunately it is so much up to us, because trust is established in the relationship… it is a relationship of trust here, almost always.>> […ecco lì la delusione… allora si deve cercare di di far comprendere che è… e da soli ovviamente non lo facciamo perché è difficilissimo, però poi tanto spetta a noi purtroppo, perché il rapporto di fiducia si instaura … è un rapporto di fiducia ecco, questo quasi sempre.] (Score: 216.460)

### 3.5. Sociodemographic Variables’ Analusis

The role of socio-demographic variables was investigated using the Chi-square test. The only variable that was significant was age [Chi2 cl2 = 5.95; *p* < 0.01]. The “young group” is more representative of the second cluster, “the adequate professional”. On the other hand, the “old group” is more representative of the fourth cluster, “the inadequate” professional”. This could indicate that younger PTs are less able to encounter the more relational and emotional part of the relationship with patients. Furthermore, owing to less experience, they may find it difficult to lay down their guard of professionalism and technique, perhaps feeling the need to demonstrate that they are valid PTs.

## 4. Discussion

Physiotherapy is an important treatment for patients suffering from different disorders. According to patient-centered care [26], one of the most important variables to consider for a good outcome is the relationship in the care process. In fact, the nature of the relationship between the PT and patient can affect the effectiveness of the treatment. An approach based on a biopsychosocial perspective, which takes into account the patient as a whole [27], could possibly explain this effect if we take into account the framework of reciprocal influence in which all health care connections occur [28]. In this way, the point of view of PTs on the relationship with the patient could have an effect on the treatment or reveal critical trigger where the relationship could be improved.

This study aims to explore how PTs represent their relationship with their patients. This was done through transcription of a sample of interviews, as well as an analysis of the social representations of PTs. In this section, we will discuss the results of the clusters, as they could be considered the cores of affection in this research.

Among the four clusters, interesting themes can be identified. The first cluster, “request of integration”, shows an integrated point of view, and it is likely that PTs deeply understand the importance of other facets besides that of the body. Relationships, emotions, and personal events can all influence the health status and the final outcome of their work, so it is important to consider these elements. This result is in line with the current literature on an integrated approach based on a biopsychosocial (BPS) vision of the patient in physiotherapy. In particular, a BPS approach led to better results in the treatment of subacute low back pain [29], chronic low back pain [30], and neck and shoulder pain [31]. This result highlights the process of interiorization of the need for a complete vision of the patient to obtain better results.

Clusters 2 (“the adequate professional”) and 4 (“the non-adequate professional”) are two clusters that stand apart from the overall context when looking at the cluster positions in the factorial space (Table 5). These two representations can define two “idealized” facets of the work, as the demon/bad and the angel/good. These representations can be easily detected in healthcare workers, as they are usually the product of a system of psychological defenses in the hard-working context [17]. These opposite areas could be the subject of a work of integration within a psychological intervention. The third cluster, the “relationship’s elaboration”, talks about the elaboration of the patients’ experience, related to their relationship with the physiotherapist. It seems possible to sustain that the themes emerging in this cluster are strongly related to the PT’s interior world and their relationship with the patient. What emerged in this cluster was the PTs’ difficulties related to elaborating on the patient’s broader life situations, which can appear difficult to sustain. Furthermore, the PT seems to try to understand what can be done with this information. This is an interesting result, as current literature is focusing on this issue, for example, proposing to teach psychological skills to PTs [32]. Finally, the socio-demographic factors allow us to emphasize the various needs of PTs in relation to their level of seniority and work experience. In this light, younger PTs could benefit from support regarding how to approach the relationship with patients in a less technical way, using some personal facets that can help build trust in patients. On a clinical note, we would like to report an interesting result regarding the specific situation that emerged after performing the interviews. In fact, after having completed the interview, many PTs thanked the interviewer for the opportunity to think about and reformulate the relationship with their patients in their work. This event appears to be clinically relevant because it may underline a PTs’ need for a space where they can reason on their way of working with their patients, through the support of a psychologist. Finally, according to a huge amount of literature, the use of narratives through interview results is functional in studying representations, especially in a relationship context. According to the present findings, as well as to the clinical evidence, these findings lead to support for a BPS approach to promote a patient-centered vision aimed to improve treatment outcomes. However, in order to generate this integration, a psychological support to PTs trough specific interventions appeared needed. In fact, the presence of a psychologist in the physiotherapeutic work could help the relationship between the PT and patient, as psychologists will support the psychological facet of the care-path and, on the other hand, will perform work of integration in PTs’ experience.

In this direction, periodic multidisciplinary meeting aimed to focus on the psychological dimensions important for enhancing the PT–patients working alliance and relationship may be useful.

Several limitations should be considered in interpreting these results. First, the small sample size. Second, the PTs come from a single Italian region, which could limit the generalizability of the results. Future studies should be completed with a broader sample of PTs living in different Italian regions. Third, the interviews took place remotely via videocall owing to COVID-19 restrictions and future studies with in-person interviews should be conducted. Moreover, the explanation of the research’s objectives to the participants may have affected the PTs’ answers. In the future, it may be possible to sample additional significant sociodemographic factors, such as the PT’s area of specialty, to look at potential differences within this profession.

## 5. Conclusions

Within the context of these limitations, our findings highlight the complexity of the work of PTs, as well as their capabilities and difficulties in working with relational and emotional elements. It seemed to emerge that PTs do not feel confident in managing these non-technical elements and living ambiguous sentiments; they make an effort to empathize while maintaining their responsibility to safeguard both the patient and themselves. Actually, physiotherapy and psychology professionals are very distant, even though they share a similar goal, such as the health of a patient. However, PTs seem to be naturally oriented to an integrated view of the body–mind relationship, suggesting that creating a point of meeting between psychology and physiotherapy could be feasible and enriching. Therefore, it would appear to be critical to foster communication between psychologists and PTs in order to encourage teamwork and collaboration. Creating collaborations between professionals and structuring a place where the PTs can discuss on the relational and emotional elements that can obstacle their work with the patients appeared to be desirable, along with the broader aim of improving treatment outcomes, promoting an integrated professional support for the body and the mind.

The sharing of these results could highlight PTs’ implicit needs associated with their work with the patients that psychology can help to improve and facilitate. This may lead to the creation of specific formative opportunities or clinical collaborations between professionals with the broader aim to realize patient-centered health promotion intervention.

## Figures and Tables

**Table 1 healthcare-10-02123-t001:** Sociodemographic and anamnestic characteristics of the sample.

Sociodemographic Variables	*M*	*SD*
Age	42	12.2
Work Experience (years)	18	12
Gender	%	*N*
Male	46	23
Female	54	27
Working Place		
Structures affiliated to National Health System (NHS)	60	30
Private practice	36	18
Structures not affiliated to NHS	2	1
Cooperatives for home-intervention assistance	2	1
Patients Treated (more than one option was possible)		
Orthopedic	86	43
Neurological	62	31
Pneumological	14	7
Oncological	12	6
Cardiological	10	5

**Table 2 healthcare-10-02123-t002:** Factor 1 emerged from the analysis. Keywords are presented by order of percentage of absolute contribution (CA%).

Factor 1 Work with Patient			
(Pole -) Visible Relationship	CA%	(Pole +) PT’s Internal Predisposition	CA%
Bond (rapporto)	5.37	Person (persona)	5.22
Rehabilitation (riabilitazione)	3.03	To talk (parlare)	3.39
To create (creare)	2.76	To Understand (capire)	3.16
Patient (paziente)	1.67	To See (vedere)	2.73
Trust (fiducia)	1.67	To Search for (cercare)	2.83
To Change (cambiare)	1.63	To Find (trovare)	2.27
Relationship (relazione)	1.21	To Put (mettere)	1.75
Reality (realtà)	1.2	To Feel (sentire)	1.52
Pathology (patologia)	1.09	To Become (diventare)	1.47

**Table 3 healthcare-10-02123-t003:** Factor 2 emerged from the analysis. Keywords are presented by order of percentage of absolute contribution (CA%).

Factor 2 The Healing Process			
(Pole -) Pathology (Patologia)	CA%	(Pole +) Relationship (Relazione)	CA%
Pathology (patologia)	5.03	Bond (rapporto)	10.52
Problem (Problema)	4.56	To create (creare)	6.1
To Change (cambiare)	2.96	To Rehabilitate (riabilitare)	5.33
Lavoro (job)	2.17	Trust (fiducia)	2.24
Neurological (neurologico)	1.93	To Establish (instaurare)	1.97
Orthopedic (Ortopedico)	1.66	To think (pensare)	1.72
Series (Serie)	1.53	In_mind (in_mente)	0.99
Field (Ambito)	1.51	Possible (possibile)	0.88
Sense (Senso)	1.44	Excellent (ottimo)	0.69

**Table 4 healthcare-10-02123-t004:** Factor 3 emerged from the analysis. Keywords are presented by order of percentage of absolute contribution (CA%).

	Factor 3 PT as a Psychologist		
(Pole -) Comprehension (comprensione)	CA%	(Pole +) Action (azione)	CA%
To see (Vedere)	4.4	To understand (capire)	12.02
To feel (sentire)	4.02	Need (Bisogno)	8.08
To Put (mettere)	2.91	To Search for (cercare)	7.4
To Find (trovare)	2.91	To Help (aiutare)	5.62
To Talk (parlare)	2.45	To Listen (ascoltare)	3.14
Life (Vita)	1.52	Person (persona)	1.08
To happen (Capitare)	1.11	Body (corpo)	1.07
To tell (raccontare)	1.05	Mind (mente)	1.06
Interesting (interessante)	1.01	Method (approccio)	0.8

**Table 5 healthcare-10-02123-t005:** Cluster location in the symbolic space.

Cl	Factor 1	Factor 2	Factor 3
	Work with Patient	The Healing Process	PT as a Psychologist
1	PT’s Internal Predisposition	Relationship	Action
	(0.38)	(0.01)	(0.49)
2	Visible relationship	Pathology	Comprehension
	(−0.44)	(−0.61)	(−0.01)
3	PT’s Internal Predisposition	Relationship	Comprehension
	(0.42)	(0.02)	(−0.48)
4	Visible Relationship	Relationship	Comprehension
	(−0.51)	(0.53)	(−0.01)

**Table 6 healthcare-10-02123-t006:** Clusters emerged from the analysis (the original Italian keywords are shown in parentheses).

Cluster	UC	Label	Keyword
1	26.7%	Request of integration	Person (*persona*) To search for (*cercare*) To understand (c*apire*) Need (*bisogno*) To help (*aiutare*) Be able to (*riuscire*) To listen (*ascoltare*) Method (*approccio*) Emotions (*emozioni*) Body (*corpo*)
2	22.2%	The adequate professional	Patient (*paziente*) Job (*lavoro*) Problem (*problema*) Pathology (*patologia*) To Change (*cambiare*) To Work (*lavorare*) Different (*diverso*) Psichology (*psicologia*) Family (*famiglia*) Psychologist (*psicologo*)
3	27.3%	Relationship’s elaboration	Person (*persona*) To See (*vedere*) To Talk (*parlare*) To find (*trivare*) To put (*mettere*) To feel (*sentire*) Life (*vita*) Situation (*situazione*) To become (*diventare*) To happen (*capitare*)
4	23.8%	The “not-adequate” professional	Patient (*paziente*) Bond (*rapporto*) To create (*creare*) Rehabilitation (*riabilitazione*) To Think (*pensare*) To live (v*ivere*) Trust (*fiducia*) To Enter (*entrare*) Pain (*dolore*) Relationship (*relazione*)

## Data Availability

The data presented in this study are available on request from the corresponding author.

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
