# Peer review of "The Relationship between Physiotherapist and Patient: A Qualitative Study on Physiotherapists’ Representations on This Theme"

_healthcare, 2022, doi:10.3390/healthcare10112123_

Round 1
Reviewer 1 Report
Additional study along these lines might consider development of the PT-patient relationship over time and the possible changing effectiveness of physiotherapy during a course of treatments or multiple courses of treatments of a patient by a PT. Is the effectiveness of treatment enhanced if the PT-patient relationship develops well over time?
One suggested correction. The Abstract mentions the 27 female PTs but not the 23 male PTs (line 12), unlike the description of participants (lines 97-98).
Author Response
Additional study along these lines might consider development of the PT-patient relationship over time and the possible changing effectiveness of physiotherapy during a course of treatments or multiple courses of treatments of a patient by a PT. Is the effectiveness of treatment enhanced if the PT-patient relationship develops well over time?
Authors: We thank the reviewer for the positive evaluation and for the suggestion. We deepened this point in the introduction reporting the results of a scoping review on the effect of therapeutic alliance in promoting treatment outcome.
“During time, studies aiming to improve the effectiveness of physiotherapy treatment included the investigation of several mediators as communication, psychological interactions between patients and clinicians and, more specifically, the therapeutic alliance. Therapeutic alliance refers to the relational processes established between therapist and client through collaboration, communication, therapist empathy, and mutual respect (Cole & McLean, 2003; Babatunde et al., 2017) that can act in combination or independently of specific treatments. The current evidences sustaining the positive effect of the therapeutic alliance in enhancing treatment outcome in health care conditions arises largely from psychotherapy and medicine literature (Babatunde et al., 2017). Babatunde et al (2017) in their scoping review found that therapeutic alliance has been poor studied in the physiotherapy rehabilitation literature finding that it can enhance a better exercise adherence. They concluded that promote the therapeutic alliance appear important for improving physical rehabilitation programs and further studies to deepen this theme, especially from therapist prospective, are needed. In this light…”
The following references have been added:
Babatunde F, MacDermid J, MacIntyre N. Characteristics of therapeutic alliance in musculoskeletal physiotherapy and occupational therapy practice: a scoping review of the literature. BMC Health Serv Res. 2017 May 30;17(1):375. doi: 10.1186/s12913-017-2311-3
Cole MB, McLean V. Therapeutic relationship re-defined. Occup Ther Mental Health. 2003;19:33–56.
One suggested correction. The Abstract mentions the 27 female PTs but not the 23 male PTs (line 12), unlike the description of participants (lines 97-98).
Authors: Thank you for your suggestions, according to this request the number of male participants has been added in the abstract section.

Reviewer 2 Report
Authors need to justify the inclusion of a physiotherapist with two years of experience. Why not a fresher physiotherapist.
Please include a paragraph for sample size calculation if performed.
Please also explain which texts were analyzed by the T-LAB software. If needed, details can be provided as a separate appendix.
Line No. 103, Why did the author explain the method of study to the participants? This might have influenced the response from the physiotherapist.
Did the author follow guidelines to conduct a structured interview? If this is the case, please provide a reference.
Line no. 114 that says texts (yes/not) does not seem right.
Line no.117, who conducted the interview and what was his/her experience.
Line no. 125, is it appropriate to write PTs representation or PTs prospective.
Line 126, please justify why the Emotional Text Mining methodology was used in this study. Please quote the validity and reliability of the method.
Line 146-150, please elaborate the section as statistical analysis.
Line no. 149, please add zero and a space before the decimal. (p < 0.05)
Results
Please group the identified factors into paragraphs with some separate headings, such as factor 1 (name of the factor).
Please provide a demographic data table of the PTs included in the study. Please also provide their specialization, qualifications, and experience. There is a vast difference between the different specializations of PT in terms of patient-therapist interaction. We would like to get a response from those PTs who actually spend time with patients rather than just consulting.
Discussion
The discussion is very short and does not significantly discuss the results of the study with reference to other available literature. Please compare and contrast other available research (other medical field research) with your results and also provide your own inference.
Conclusion
Please provide a brief and clear conclusion. What was drawn from the study and how will this knowledge be used among PT community.
Author Response
Authors need to justify the inclusion of a physiotherapist with two years of experience. Why not a fresher physiotherapist.
Authors: Thank you for this comment. We decided to include participant with at least two years of experience for the following reasons: 1) we aimed to collect information of professionals with experience in working with patients not at the very beginning of their work 2) we aimed to collect information from professional that was working even before the COVID-19 pandemic and that have experience not totally affected by this difficult period.
We briefly have added this information in the inclusion criteria.
“3) working as a PT for at least two years considering two years the minimum needed to collect information from professionals with working experience also before COVID-19 pandemic outbreak.”
Please include a paragraph for sample size calculation if performed.
Authors: Thank you for this comment. We did not perform a classic analysis which allows us to make a sample size calculation. However, according to the text mining analysis, we calculated the lexical indexed of the final corpus (the whole sample of interviews) to assess if the final corpus was enough rich to proceed with the analysis (lines 148-149). Now we added these indexes as follow in Line 173:
“The lexical indexes of the final corpus (TTR: 0.089; HAPAX%: 52%) assessed its richness and allowed us to continue the analysis according to ETM methodology”
Moreover, we specified these indexes in the previous section in line 149-151.
Please also explain which texts were analyzed by the T-LAB software. If needed, details can be provided as a separate appendix.
Authors: Thank you for your suggestion. T-LAB software analyzes the transcription of PTs’ interviews, we tried to specify this in the first part of results (Line 172):
“The transcription verbatim of all 50 interviews created a whole corpus of medium dimension (Token:74763).”
Line No. 103, Why did the author explain the method of study to the participants? This might have influenced the response from the physiotherapist.
Authors: Thank you for this comment. We rephrase this point since as it was written was confounding. In fact, the psychologist explained only that the study was aimed to collect information about the relationship PT-patients through a telematics interview. We modified the sentence as follow:
More specifically, the psychologist informed the PTs that we were conducting a study aimed to collect personal considerations about the relationship between PT and their patients through a telematics interview
Did the author follow guidelines to conduct a structured interview? If this is the case, please provide a reference.
Authors: Thank you for you suggestion. We used the guidelines of the manual referenced as Corbetta 2009 [11] as specified in Line 136-138
Line no. 114 that says texts (yes/not) does not seem right.
Authors: As requested we modify the sentence “(…) performing home interventions or not performing home interventions (…).
Line no.117, who conducted the interview and what was his/her experience.
Authors: Thank you for this comment. We added this information in the Procedure section.
“All interviews were conducted by two psychologists with experience in the conduction of interview”.
Line no. 125, is it appropriate to write PTs representation or PTs prospective.
Authors: According to the social psychology and to the theory of social representation, we preferred representations (see reference 16, Greco, 2016).
Line 126, please justify why the Emotional Text Mining methodology was used in this study. Please quote the validity and reliability of the method.
Authors: Thank you for your comment, we improved this part as follows:
“Such methodology of Natural Language Processing (NPL) is based on a socio-constructivist approach that, based on the association of words, and it allows to identify the general themes and the symbolic-cultural categories of construction of meaning.
And
“Moreover, it has been already applied to study patients experiences in a patient-centered view [19].”
For the reliability and validity of the method, we added this part:
“To deepening the validity and the reliability of this methodology, please see [14, 20, 21, 22].”
Line 146-150, please elaborate the section as statistical analysis.
Author: According to this suggestion the last paragraph of the data analysis was insert in a new section named Statistical Analysis.
“2.4 Statistical Analysis
Through the Statistical Package for Social Science version 25 for Windows (SPSS version 25; IBM, Armonk, NY, USA) we performed a Chi-square test on the final partition of the Cluster, following the socio-demographic variables collected. This way, we can identify the significant presence of a variable in a Cluster. A p < 0.05 was considered significant.”
Line no. 149, please add zero and a space before the decimal. (p < 0.05)
Authors: As requested both a zero and a space have been added before the decimal.
Results
Please group the identified factors into paragraphs with some separate headings, such as factor 1 (name of the factor).
Authors: Thank you, we added the paragraphs for each factor. Moreover, we added a paragraph also for the cluster results (3.5).
Please provide a demographic data table of the PTs included in the study. Please also provide their specialization, qualifications, and experience. There is a vast difference between the different specializations of PT in terms of patient-therapist interaction. We would like to get a response from those PTs who actually spend time with patients rather than just consulting.
Authors: Thank you for your suggestions, we provided a new table with the important information that You suggested. As variable from the sociodemographic questionnaire, we did not collect the specialization of PTs but only the kind of patient that they are used to treat (orthopedic, neurological…). We added this in the new Table 1 in the Participants section.
Discussion
The discussion is very short and does not significantly discuss the results of the study with reference to other available literature. Please compare and contrast other available research (other medical field research) with your results and also provide your own inference.
Authors: Thank you, we improved this section following your suggestions.
Conclusion
Please provide a brief and clear conclusion. What was drawn from the study and how will this knowledge be used among PT community.
Authors: Thank you, we improved the Conclusions following your suggestion.

Round 2
Reviewer 2 Report
Dear Authors,
Thanks a lot for addressing the comments. I have a minor comment on the limitations section of the study.
1.Please include limitations such as small sample size and generalizability constraints.
2. The limitations of the procedure, which I mentioned in my previous review.
Additionally, please rephrase the few complex and passive sentences in the introduction and discussion.
Author Response
Dear Reviewer,
we already included a paragraph on Limitations including the small sample and generalizability (see line 430).
We deepened this section including other limitations (see line 435) that you previously suggested.
Moreover, we have done a rephrasing of some sentences (e.g. Line 374; Line 393; Line 407; Line 61) as you suggested
